# Differential Contribution of Protein Factors and 70S Ribosome to Elongation

**DOI:** 10.3390/ijms22179614

**Published:** 2021-09-05

**Authors:** Alena Paleskava, Elena M. Maksimova, Daria S. Vinogradova, Pavel S. Kasatsky, Stanislav V. Kirillov, Andrey L. Konevega

**Affiliations:** 1Petersburg Nuclear Physics Institute, NRC “Kurchatov Institute”, 188300 Gatchina, Russia; polesskova_ev@pnpi.nrcki.ru (A.P.); elena.maks.89@gmail.com (E.M.M.); vinogradova_ds@pnpi.nrcki.ru (D.S.V.); Kasatskiy_PS@pnpi.nrcki.ru (P.S.K.); kirillov_sv@pnpi.nrcki.ru (S.V.K.); 2Institute of Biomedical Systems and Biotechnology, Peter the Great St. Petersburg Polytechnic University, 195251 St. Petersburg, Russia; 3NRC “Kurchatov Institute”, 123182 Moscow, Russia

**Keywords:** translation, elongation factor, 70S ribosome, rapid kinetics, heterologous system, antibiotics

## Abstract

The growth of the polypeptide chain occurs due to the fast and coordinated work of the ribosome and protein elongation factors, EF-Tu and EF-G. However, the exact contribution of each of these components in the overall balance of translation kinetics remains not fully understood. We created an in vitro translation system *Escherichia coli* replacing either elongation factor with heterologous thermophilic protein from *Thermus thermophilus*. The rates of the A-site binding and decoding reactions decreased an order of magnitude in the presence of thermophilic EF-Tu, indicating that the kinetics of aminoacyl-tRNA delivery depends on the properties of the elongation factor. On the contrary, thermophilic EF-G demonstrated the same translocation kinetics as a mesophilic protein. Effects of translocation inhibitors (spectinomycin, hygromycin B, viomycin and streptomycin) were also similar for both proteins. Thus, the process of translocation largely relies on the interaction of tRNAs and the ribosome and can be efficiently catalysed by thermophilic EF-G even at suboptimal temperatures.

## 1. Introduction

The ribosome is a macromolecular complex that translates the genetic information into the amino acid sequence of proteins. Although the ribosome is the platform and the central component of protein biosynthesis, the protein elongation factors Tu and G (EF-Tu and EF-G) are crucial in accelerating and increasing the accuracy of elongation. EF-Tu delivers aminoacylated tRNA (aa-tRNA) to the aminoacyl (A) site of the ribosome. The aa-tRNA recognises mRNA codon and ensures insertion of correct amino acid into a growing polypeptide chain. EF-G catalyses translocation reaction—the coordinated movement of two tRNAs and the mRNA. As a result, the new mRNA codon is located in the vacant A site, making the ribosome ready for the next cycle of elongation. Key features of the functioning of these components of the translational apparatus are known, but the details of these processes remain not fully understood. Moreover, different methods, reaction conditions and different model organisms do not always allow the assembly of scattered puzzle pieces. Thus, the most of genetic and biochemical data revealing the peculiarities of the functioning of translation refer to mesothermal microorganism *Escherichia coli* (*E. coli*), while structural interpretation of protein biosynthesis primarily relates to the data obtained using components of thermophilic microorganisms, in particular, *Thermus thermophilus* (*T. thermophilus*). The comparability of these data seems to be not entirely unambiguous, given the enormous evolutionary distance between *E. coli* and *T. thermophilus* [1] and significantly different habitat conditions of these microorganisms. However, the results of *T. thermophilus* completed genome sequencing showed that the translational factors of this thermophilic microorganism are homologous to those of *E. coli* [2]. The sequence identity of EF-Tu from *T. thermophilus* and *E. coli* is 71%, 82% of all amino acids of these proteins show similar chemical properties; for EF-G, these values are 60% and 74% [3]. Spatial structures of thermophilic and mesophilic elongation factors interacting with guanine nucleotides, aa-tRNA or the ribosome depict significant resemblance [4,5,6,7,8,9]. Elongation factors from *E. coli* and *T. thermophilus* also exhibit similar functional properties. The cooperative participation of all three EF-Tu domains is necessary to ensure a phenotype regarding the binding of guanine nucleotides both in *T. thermophilus* [10] and *E. coli* [11]. While, for example, high affinity for GDP and GTP is an inherent property of the G-domain in EF-Tu from *Bacillus stearothermophilus* [12], and partial truncation of *Sulfolobus solfataricus* EF-1α increases the binding affinity of GDP and GTP by approximately one order of magnitude compared to intact protein [13]. Mutational analysis of catalytic His84 in *E. coli* EF-Tu and homologous His85 in *T. thermophilus* EF-Tu revealed similar mechanisms of GTP hydrolysis by these two proteins [14,15,16]. Studying the thermodynamic parameters of aa-tRNA binding to *E. coli* and *T. thermophilus* EF-Tu unveiled considerable similarity [17]. Thermophilic EF-Tu supported mesophilic protein biosynthesis both in vivo and in vitro [18], albeit with reduced efficiency of 30% [19]. The functional comparison of EF-G from *E. coli* and *T. thermophilus* is not so widely presented in the literature. A search revealed that EF-G from *T. thermophilus* added to the mesophilic system hydrolyses GTP and catalyses the translocation of tRNA, such as homologous protein, but at a significantly lower rate [20]. Thus, *E. coli* and *T. thermophilus* elongation factors exhibit a high degree of similarity in various aspects and can substitute each other in a heterologous system [19,21].

In this work, *E. coli* reconstituted in vitro translation system with the elongation factors substituted for those from the thermophilic microorganism *T. thermophilus* was used to study two fundamental reactions of the elongation cycle: the A-site binding of aa-tRNA and the displacement of peptidyl-tRNA from the A to the peptidyl (P) site of the ribosome during translocation. This approach provides a biochemical characterisation of individual thermophilic components in the context of certain partial translation reactions. It also provides a deeper understanding of the mesophilic system’s functioning and merges extensive structural information to biochemical and genetic data.

## 2. Results and Discussion

### 2.1. Kinetics of tRNA Interaction with the Ribosome Is Determined by the Properties of Elongation Factor EF-Tu

Delivery of the tRNA-bound amino acid to the peptidyl transferase centre of the ribosome is a multistep process. The application of a large arsenal of methods, reporter groups and modified components made it possible to identify and describe details of the elementary steps [22]. To compare the contribution of homologous and heterologous elongation factors EF-Tu, we decided to monitor a relatively late event of the A-site binding—tRNA accommodation—that would allow us to detect potential changes caused by heterologous EF-Tu during accommodation or preceding steps. The aa-tRNA accommodation in the A site changes the fluorescence intensity of the proflavin label (Prf) located at the elbow region of the initiator tRNA(Prf20) in the P site of the ribosome [23,24]. The interaction of both ternary complexes EF-Tu∙GTP∙Phe-tRNA^Phe^ with the ribosome resulted in a fluorescence intensity decrease (Figure 1a). The values of the apparent rate constants increased hyperbolically with an increase in the concentration of the ternary complex with the maximal velocity of the accommodation reaction 6.9 ± 0.7 s^−1^ (*E. coli*) and 0.76 ± 0.04 s^−1^ (*T. thermophilus*) at a temperature of 20 °C, 35 ± 3 s^−1^ (*E. coli*) and 3.8 ± 0.3 s^−1^ (*T. thermophilus*) at 37 °C (Figure 1b). Thus, at 20–37 °C, thermophilic EF-Tu reduces the rate of partial reactions in the A site of the ribosome by one order of magnitude compared to the mesophilic elongation factor. These data support the earlier notion of lowered efficiency of polypeptide synthesis in the case of substitution of *E. coli* EF-Tu by one from *T. thermophilus* in the in vitro reconstructed system [19]. Several aspects could be at the core of such a drop in the functioning efficiency of the thermophilic factor. The most obvious rationale is the fact that temperature in the range of 20–37 °C is significantly lower than both the minimum growth (47 °C) and the optimum functioning temperature (73 °C) of *T. thermophilus* [25]. Homologous protein was ‘optimised’ during evolution to function with this particular system, whereas minor alterations in the amino acid sequences and conformations of EF-Tu from two organisms could lead to a rate discrepancy of one or several steps at the stage of tRNA delivery, despite the overall structural similarity.

### 2.2. Tetracycline Interacts with the Ribosome and Kinetically Inhibits tRNA Binding to the A Site

Antimicrobial agents that inhibit the delivery and accommodation of aa-tRNA on the ribosome provide additional information about the details of partial reactions in the A site. Tetracycline binds to the small ribosomal subunit and sterically violates the recognition of an mRNA codon by the aa-tRNA [26]. High conservation of its binding site on bacterial and eukaryotic ribosomes [27] suggests that the mechanism of inhibition is characteristic for many organisms and causes a wide spectrum of tetracycline action. Since tetracycline binding to the ribosome leads to a dramatic decrease in the rate of A-site binding [28], Prf-labelled tRNA cannot be used as a fluorescence reporter due to photobleaching of the dye. Instead, we have used the signal of the BODIPY FL (BPY) label attached to the acceptor end of the initiator tRNA in the P site (Appendix A) that remains stable at least for an hour. This reporter group was previously used in a Föster Resonance Energy Transfer (FRET) pair for analysing the kinetics of A-site binding of tRNA [29] that inspired us to test direct fluorescence signals. The signal was attributed to the relatively late event of the A-site binding, presumably accommodation, as no characteristic fluorescence increase was detected for the reactions up to dissociation of EF-Tu after GTP hydrolysis (with the use of a modified variant of EF-Tu His84Ala deficient on GTP hydrolysis [14,15] and kirromycin-stalled ribosomal complexes [14,30]) (Figure 2). Since the time courses had two exponential terms in the opposite directions (Appendix A), for a better comparison with the Prf20 signal, we analysed average rate constants, *k_av_*, corresponding to the observed reaction half-life time, rather than *k_app_* values of individual exponentials. The rates at the saturation level (9.6 ± 0.8 s^−1^ (*E. coli*), 0.74 ± 0.02 s^−1^ (*T. thermophilus*) at 20 °C and 29 ± 3 s^−1^ (*E. coli*), 2.9 ± 0.2 s^−1^ (*T. thermophilus*) at 37 °C) (Appendix A) were comparable to those obtained with Prf20 fluorescence signal.

The addition of 30 μM tetracycline led to a dramatic decrease in the rate of A-site binding of tRNA, both in the case of thermophilic and mesophilic elongation factors (0.0019 ± 0.0002 s^−1^ and 0.0024 ± 0.0001 s^−1^, respectively) (Figure 2). The final level of inhibition was almost the same for both complexes, despite differences in the A-site reaction rates without the antibiotic. The amplitude of the fluorescent signal with antibiotics was somewhat reduced but remained comparable to the signal obtained in the absence of an inhibitor. The results emphasise that we are dealing with the ribosome-mediated action of tetracycline: the antibiotic binds to the ribosomal complex and leads to kinetic inhibition of the A-site reaction, not depending on the nature of the elongation factor used.

### 2.3. Species Differences of Elongation Factor EF-G Do Not Influence Translocation Kinetics

Upon translocation, the movement of the acceptor end of tRNA and mRNA (and hence the anticodon of the peptidyl-tRNA) is synchronised and occurs universally fast [31]. However, interactions of the acceptor and anticodon parts of tRNA with the ribosome are of great importance when tRNA is bound to classical ribosome sites [32], while contacts with the central part of tRNA—elbow of L-shaped structure—become essential when tRNA moves between sites [33,34]. In this work, we evaluated the rates of movement of elbow and acceptor parts of peptidyl-tRNA molecule as it travels from the A to the P site of the ribosome during GTP-dependent EF-G catalysed translocation. Fluorescence change in proflavin label located in the D-loop of fMet-Phe-tRNA^Phe^(Prf16/17) accompanied elbow movement, while the signal of BPY label attached to methionine of BPY-Met-Phe-tRNA^Phe^ reflected the movement of the acceptor CCA-end of tRNA in the peptidyl transferase centre of the large ribosomal subunit (Figure 3). In both cases, the addition of EF-G to the pretranslocation complex resulted in a two-step change in fluorescence intensity with a predominant by amplitude (85–90% of the total signal amplitude) fast step that was analysed further unless stated otherwise.

The velocity of the movement of the elbow region of tRNA was comparable for thermophilic and mesophilic factors in the temperature range from 15 to 42 °C (Figure 4a, Table 1). However, at a higher temperature exceeding the functioning optimum for the mesophilic factor (50 °C), the rate of the reaction catalysed by the thermophilic factor was higher, presumably due to partial denaturation of the mesophilic factor. The movement of the acceptor end of tRNA had a slight trend towards an increase of difference in the observed reaction rates with decreasing temperature (Figure 4b, Table 1).

In summary, one can say that in the case of the mesophilic factor, the dependence of the velocity of the tRNA elbow movement on the temperature is more pronounced than that of the acceptor end of tRNA: at lower temperatures, the elbow moves slower than the acceptor end, and at elevated temperatures—faster (Figure 4c). The rate of movement of both parts of tRNA coincides approximately at 33 °C. At higher temperatures, the rate of elbow movement exceeds the rate of tRNA acceptor end. During translocation with the thermophilic EF-G, the values of the velocity of movement of both parts of tRNA are similar at 15 °C, whereas in the rest of the temperature range studied, the relocation of the elbow of tRNA noticeably outpaces the movement of the acceptor end of tRNA, with maximum differences reached at 42 and 50 °C (Figure 4d).

Thus, the tRNA elbow’s movement precedes the movement of the acceptor end to the post-translocation state. Relocation of the central part of tRNA does not depend on the nature of the elongation factor, highlighting the role of interactions between the ribosome and tRNA. Either EF-G can create favourable conditions for this reaction, fulfilling the pre-conditioning of the ribosomal complex. The tRNA acceptor end movement revealed some dependence on the elongation factor: replacing homologous EF-G with thermophilic protein decreased the reaction rate. The most pronounced effect occurred at a lower temperature (maximum two times at 15 °C), the least acceptable for the effective functioning of the thermophilic translocation catalyst. This observation emphasises that the movement of the acceptor end of tRNA in the time range of the study is not spontaneous but is induced by (and depends on) the elongation factor. This result is consistent with the translocation model that separates the movement of the tRNA CCA-end to the post-translocation position and hybrid state [35]. The tRNA shift between classical and hybrid states is due to spontaneous ribosome subunit rotations [35]. These transitions reflect internal conformational dynamics of the ribosome and are not kinetically related to EF-G binding or tRNA translocation [31]. We believe that tRNA translocation, observed in this work, is mainly determined by varying contacts with the ribosome, whereas EF-G provides an appropriate environment for the reaction. Modest impairment caused by thermophilic EF-G on CCA-end movement without comparable effect on the central part of tRNA emerges as the relocation of the acceptor end of tRNA is a low energy-consuming process. The impact noticeable for the movement of CCA region of tRNA, which forms a small number of contacts, ceases to be visible in the case of more global conformational changes—the displacement of the tRNA elbow.

In support of this assumption, we determined the activation energy of translocation by constructing Arrhenius plots for the rate of translocation reaction (Figure 5), using different fluorescent labels at different temperatures. The activation energy determined by the label at the elbow region of tRNA coincided in systems with mesophilic and thermophilic elongation factors (81 ± 4 kJ/mol and 86 ± 4 kJ/mol, respectively) and exceeded the values obtained for the reactions monitored by the labels at the acceptor ends of tRNA (57 ± 1 kJ/mol in the presence of EF-G from *E. coli* and 70 ± 3 kJ/mol with the participation of thermophilic EF-G). The values of the activation energy are under previously obtained data for translocation in the *E. coli* system in the presence of EF-G·GTP (68 kJ/mol) [36], which coincides with the average value of the activation energy of translocation monitored at the acceptor and elbow regions of tRNA (69 kJ/mol).

### 2.4. Antibiotics Bind to the Small Ribosomal Subunit and Inhibit the Movement of the Elbow Region of tRNA

Multistep translocation process—binding of EF-G, GTP hydrolysis, movement of two tRNAs and the mRNA, dissociation of the factor and tRNA from the E site—takes place in milliseconds. The detailed study requires inhibitors to slow down or block the process at a particular step. Due to heterogeneity of elongation factors, no compounds directly affecting the functioning of EF-G, for example, disturbing accommodation (thiopeptides) or inhibiting dissociation (fusidic acid) of the factor, were used. We considered a group of antibiotics interacting with the small ribosomal subunit, with the main influence on the movement of tRNA: streptomycin, viomycin, hygromycin B and spectinomycin.

Streptomycin is one of those antibiotics that can be assigned to several specific categories of elongation cycle inhibitors. It significantly reduces translation accuracy and inhibits translocation. Streptomycin binds to the small ribosomal subunit and causes a conformation that reduces the selectivity for tRNA [37] both during initial selection and at the proofreading step [38,39,40,41]. The antibiotic enhances tRNA binding affinity to the A site [38,42] that increases the likelihood of inserting incorrect amino acids, and leads to inhibition of translocation [43]. Streptomycin reduces the rate of translocation of the entire tRNA molecule (Figure 6, Table 2); however, the impact on the movement of the tRNA elbow (3–4 times decrease) is somewhat more pronounced than that of the acceptor end of tRNA (1.5–2 times decrease). Presumably, the binding of streptomycin causes the conformation of the ribosome that more heavily disturbs the movement of the central part of tRNA.

Binding of viomycin to the ribosome occurs in the intersubunit region between Helix 44 and Helix 69 of the small and large ribosomal subunits, respectively [44,45], causing stabilisation of a specific intermediate hybrid position of the ribosome [46] with the anchoring of tRNA in the A site [47]. It increases the probability of erroneous reading of the A codon with stimulation of reverse and inhibition of direct translocation, similarly to streptomycin action. However, the severity of the inhibitory effect on translocation is infinitely greater, including desynchronisation of tRNA movement (Figure 6, Table 2). The addition of 200 μM viomycin leads to an enormous decrease in the rate of tRNA elbow displacement upon translocation with thermophilic factor (more than 8000 times) and complete inhibition of this reaction in the case of a mesophilic elongation factor (within a time of an experiment 100 s). At the same time, the rate of movement of the acceptor part of tRNA increases 1.6–2 times. Thus, the conformation of the ribosomal complex, caused or stabilised by the addition of viomycin, leads to an accelerated movement of the acceptor end with complete inhibition of the movement of the central part of the tRNA with both effects being more expressed in the mesophilic system.

The next translocation inhibitor—hygromycin B—interacts with the helix 44 of the small subunit, such as viomycin, but does not cause an increased frequency of translation errors or stimulate reverse translocation. The addition of hygromycin B moderately accelerates acceptor end movement and slows the motion of the rest of the tRNA molecule (Figure 6, Table 2) with full amplitude signals from the labels in the tRNA elbow and CCA-end (current study) and extremely small—from the anticodon of tRNA [31]. Almost complete inhibition of the movement of the anticodon loop and maintaining of large-scale relocation of the central and acceptor parts of tRNA are following hygromycin B binding at the decoding centre [48]. It interacts with the 16S rRNA bases that are in contact with mRNA and tRNA, blocks the movement of these substrates between the A and the P sites and prevents both direct and reverse translocation [49].

The main inhibitory effect of spectinomycin aims in violation of the movement of tRNA and mRNA. The rigid spectinomycin molecule binds to the helix 34 in the flexible region of the head of the small subunit [37] and captures a specific rotated head conformation [49]. Restriction of the head mobility causes an inability of a full-scale move of tRNA to the P and the E sites: translocation of the codon–anticodon duplex is practically absent [31]. At the same time, transfer of the acceptor end of tRNA occurs two times faster, and the amplitude of the signal responsible for this movement 1.5 times exceeds the same parameter of translocation without the inhibitor (Figure 6, Table 2). In addition, spectinomycin increases the severity of the two-step character of the signal describing the translocation of the elbow region of tRNA. The rate of the first fast step remained almost unchanged (with a 1.5 times increase in the case of *E. coli* and no change in the case of EF-G from *T. thermophilus*), while the amplitude of the fluorescent signal of this step was significantly declined. The amplitude of the second step rose to 46% for *E. coli* and 60% for *T. thermophilus* EF-G with 20 times decrease in the rate approaching an extremely slow translocation in the presence of hygromycin B. Thus, the addition of spectinomycin makes it possible to detect the intermediate position of tRNA during translocation [50], featuring fast movement of the acceptor end of tRNA, stepwise movement of elbow region and almost complete cessation of translocation of the tRNA anticodon region.

Differential analysis of fluorescent signals showed that inhibition of GTP-dependent translocation by the set of antibiotics had similar patterns in cases of thermophilic and mesophilic factors. All antibiotics had a much more sizeable effect on the movement of the elbow compared to the acceptor end of tRNA. The binding of antibiotics to the small ribosomal subunit can logically explain their pronounced inhibitory effect on codon–anticodon duplex translocation [31], while our data show that the impact extends to more remote areas of the ribosome, most likely, partially disrupting the formation of 23S rRNA contacts with the elbow region of tRNA that are necessary for translocation of tRNA.

In summary, both thermophilic elongation factors (EF-Tu and EF-G) support mesophilic translation, albeit with varying efficiency. Dynamic interaction of EF-Tu with aminoacylated tRNA should be strong enough to deliver tRNA to the ribosome and readily weaken to release it to the A site. Thermophilic EF-Tu confuses fine-tuning of this process due to partially distinct amino acid sequence, conformation peculiarities and more pronounced structural rigidity enabling functioning at elevated temperatures. At the same time, likely, species-specific sequence differences of EF-G do not play any critical role in translocation. The overall similarity of protein structures from two organisms is sufficient to perform unaltered heterologous translocation in *E. coli* since prevailing contacts are between tRNA and the ribosome. EF-G catalyses the process mainly focused on disruption of codon–anticodon interaction of tRNA with the P site and relocation it to the E site of the ribosome. Since thermophilic factor is adapted to implement this process in a more rigid system, it may well promote the translocation in mesophiles. Thus, we suggest that homologous components are necessary for effective tRNA delivery to the A site, whereas translocation can be supported by a heterologous factor from a more thermostable organism.

## 3. Materials and Methods

### 3.1. Materials

Experiments were carried out in buffer TAKM_7_ (50 mM Tris–HCl (pH 7.5), 70 mM NH_4_Cl, 30 mM KCl, 7 mM MgCl_2_) unless otherwise stated. fMet-tRNA^fMet^ (*E. coli*) was prepared as described [51], as were BPY-Met-tRNA^fMet^ (*E. coli*) [31], tRNA^fMet^(Prf20) (*E. coli*), tRNA^Phe^(Prf16/17) (yeast) [52], initiation factors (IF1, IF2, IF3) [53], 70S ribosomes from *E. coli* MRE600 cells [54] and mRNA (71 nt, coding sequence starting with fMetPhe) [55,56]. Antibiotics were from Merck (Darmstadt, Germany) and were used at the following concentrations: tetracycline, 30 µM [24]; hygromycin B, 20 µM; viomycin, 200 µM; spectinomycin, 900 µM; streptomycin, 20 µM [31]; kirromycin, 150 µM [57].

### 3.2. Protein Expression and Purification

The plasmid containing the *E. coli tufB* gene cloned into a pET-24a vector and coding for EF-Tu (P0CE48) extended by six histidine residues at the C terminus as well as similar plasmid coding for modified variant EF-Tu His84Ala were kindly provided by Prof. Marina Rodnina (MPI BPC, Goettingen, Germany). The plasmid containing the *T. thermophilus tufB* gene cloned into a pET-3a vector and coding for EF-Tu (P60339) extended by six histidine residues at the N terminus was kindly provided by Prof. Olke Uhlenbeck (Northwestern University, Evanston, IL, USA). The plasmid containing the *E. coli fusA* gene cloned into a pCA24N vector and coding for EF-G (P0A6M8) extended by six histidine residues at the N terminus was from the ASKA library [58]. The *T. thermophilus fusA* gene coding for EF-G (P13551) was amplified by PCR (Bio-Rad T100 Thermal Cycler, Hercules, CA, USA) from genomic DNA isolated in our laboratory from *T. thermophilus* (strain ATCC 27634/HB8) using a forward primer (5′-GAAGAACATATGGCGGTCAAGGTAGAGTAC-3′) and a reverse primer (5′-CTACTAGTCGACTTGACCCTTGATGAGCTTCTC-3′). The PCR product was inserted into the pET-21a vector between NdeI and XhoI sites. The sequence of EF-G extended by six histidine residues at the C terminus was confirmed by DNA sequencing.

Proteins were expressed at 37 °C in *E. coli* BL21 (DE3) in LB medium with the addition of selective antibiotics (30 μg/mL kanamycin for the plasmid with EF-Tu protein gene from *E. coli*, 100 μg/mL ampicillin for the plasmid with EF-Tu from *T. thermophilus*, 30 µg/mL of chloramphenicol for the plasmid with EF-G from *E. coli* and 100 µg/mL of ampicillin for the plasmid with EF-G from *T. thermophilus*). Expression was induced by the addition of 1 mM IPTG when the cell culture reached an optical density of OD_600_ = 0.6. Protein purification was performed using Protino^®^ Ni-IDA (MACHEREY-NAGEL, Düren, Germany) under non-denaturing conditions in accordance with the manufacturer’s protocol. Proteins were dialysed against TAKM_7_ buffer containing 5 mM β-ME and 10% glycerol in the case of EF-Tu or 50% glycerol in the case of EF-G, concentrated, aliquoted, frozen in N_2_ and stored at −80 °C. Protein concentration was determined spectrophotometrically at an absorption wavelength of 280 nm using an extinction coefficient for EF-Tu *T.*
*thermophilus* ε = 25,900 M^−1^cm^−1^, for EF-Tu *E. coli* ε = 32,900 M^−1^cm^−1^, EF-G *T. thermophilus* ε = 49,280 M^−1^cm^−1^ and for EF-G *E. coli* ε = 66,200 M^−1^cm^−1^. The purity of proteins was more than 90% according to SDS-PAGE.

### 3.3. Biochemical Assays

For the preparation of initiation complexes, 2 µM ribosomes were incubated with 4 µM initiation factors (IF1, IF2, IF3), 12 µM mRNA, 4 µM initiator tRNA (fMet-tRNA^fMet^, fMet-tRNA^fMet^(Prf20), or BPY-Met-tRNA^fMet^), 1 mM GTP and 1 mM DTT in buffer TAKM_7_ for one hour at 37 °C. tRNA^fMet^(Prf20) was aminoacylated and formylated immediately before the initiation complex formation, as described previously [53]. Where necessary, initiation complexes were purified by gel-filtration chromatography on a BioSuite 450 HR SEC column, 7.8 × 300 mm (Waters, Milford, MA, USA) in buffer TAKM_7_.

For the preparation of ternary complex, EF-Tu·GTP·Phe-tRNA^Phe^, 8 µM EF-Tu *E. coli* (or 20 µM EF-Tu *T. thermophilus*) was incubated with 1 mM GTP, 1 mM DTT, 3 mM phosphoenol pyruvate, 0.5 mg/L of pyruvate kinase for 15 min at 37 °C, followed by addition of 4 µM Phe-tRNA^Phe^ (or Phe-tRNA^Phe^(Prf16/17)) and further incubation for 5 min. Phe-tRNA^Phe^ was preformed and purified [59], Phe-tRNA^Phe^(Prf16/17) was prepared immediately before ternary complex formation. For this, 8 μM tRNA^Phe^(Prf16/17) was incubated with 0.2 mM Phe, 3 mM ATP, 6 μM 2-mercaptoethanol, 40 nM phenylalanine-tRNA synthetase, 40 nM tRNA nucleotidyltransferase in buffer TAKM_7_ for 30 min at 37 °C.

For the preparation of pretranslocation complexes programmed with mRNA and carrying deacylated tRNA^fMet^ in the P site and fMet-Phe-tRNA^Phe^ in the A site, the ternary complex was added to the initiation complex in a ratio 2:1 and incubated for one minute at 25 °C. Then, the concentration of Mg^2+^ was adjusted to 21 mM, and the pretranslocation complexes were purified by ultracentrifugation through 500 mL of a 1.1 M sucrose cushion in buffer TAKM_21_ for 3 h at 4 °C at 50,000 rpm in a Optima L-90K Ultracentrifuge with SW55Ti rotor (Beckman Coulter, Brea, CA, USA). The precipitate was dissolved in buffer TAKM_21_, aliquots were frozen in N_2_ and stored at −80 °C.

### 3.4. Rapid Kinetics

Fluorescence experiments were carried out using an SX-20 stopped-flow apparatus (Applied Photophysics, Leatherhead, UK). Proflavin fluorescence was excited at 460 nm, BODIPY FL fluorescence was excited at 470 nm and measured after passing a cut-off filter KV490 nm (Schott, Mainz, Germany) in both cases. Samples were rapidly mixed in equal volumes. Time courses depicted in the figures were obtained by averaging 5–7 individual transients. Data were evaluated by fitting to a single-exponential function with a characteristic apparent rate constant (k_app_), amplitude (A), and final signal amplitude (F_∞_) according to equation F = F_∞_ + A × exp(−k_app_ × t), where F is the fluorescence at time t. Where necessary, two exponential terms were used with two characteristic apparent rate constants (k_app1_, k_app2_), amplitudes (A_1_, A_2_), according to equation F = F_∞_ + A_1_ × exp(−k_app1_ × t) + A_2_ × exp(−k_app2_ × t). Ternary complex titrations were fitted to hyperbolic functions. The average rate constants (k_av_) represented the reciprocal of the time at which the reaction reached 50% completion. Calculations were performed using Prism 6.02 software (GraphPad Software, San Diego, CA, USA). Average k_app_ values, k_av_ values and standard error of the mean (s.e.m.) were obtained using the same software with 5–7 time courses for each concentration point. All experiments were repeated 2–4 times.

To study kinetics of A-site binding, initiation complex (0.05 μM) containing fMet-tRNA^fMet^(Prf20) (initiator tRNA labelled with proflavine at position 20 in the elbow region of tRNA) [23,24] was rapidly mixed with increasing concentration of ternary complex EF-Tu∙GTP∙Phe-tRNA^Phe^ (0.1 μM, 0.2 μM, 0.4 μM, 0.8 μM, 1.2 μM, 2 μM) containing mesophilic or thermophilic EF-Tu. The reaction was monitored at 20 and 37 °C.

To study inhibition of A-site binding, initiation complex (0.1 μM) containing BPY-Met-tRNA^fMet^ (initiator tRNA labelled with BODIPY FL at methionine moiety of tRNA) was rapidly mixed with ternary complex EF-Tu∙GTP∙Phe-tRNA^Phe^ (1 μM) containing either mesophilic or thermophilic EF-Tu. The reaction was monitored at 20 °C. Where necessary, 30 μM tetracycline or 150 μM kirromycin was added. In addition, reaction was performed with a ternary complex containing EF-Tu His84Ala from *E. coli*.

To ensure comparability of results obtained with two types of fluorescence labels, we performed titration of initiation complex (0.1 μM) containing BPY-Met-tRNA^fMet^ with increasing concentration of ternary complex EF-Tu∙GTP∙Phe-tRNA^Phe^ (0.2 μM, 0.5 μM, 1 μM, 2 μM) containing mesophilic or thermophilic EF-Tu. The reaction was monitored at 20 and 37 °C.

To study the kinetics of translocation, we used pretranslocation complexes containing deacylated tRNA^fMet^ at the P site and fluorescently labelled fMet-Phe-tRNA^Phe^ at the A site. To characterise the movement of the central part of tRNA upon translocation, we used proflavine attached to dihydrouridine at positions 16 and/or 17 located in the elbow region of fMet-Phe-tRNA^Phe^ (fMet-Phe-tRNA^Phe^(Prf16/17)) [60]. To monitor displacement of the acceptor end of tRNA from the A to the P site, we used Met-Phe-tRNA^Phe^ labelled with BODIPY FL at methionine moiety (BPY-Met-Phe-tRNA^Phe^) [31]. Pretranslocation complexes (0.06 µM) were rapidly mixed with saturating concentration of EF-G (5 µM) from *E. coli* or *T. thermophilus* in the presence of 1 mM GTP. The reaction was monitored at different temperatures (15, 20, 25, 30, 37, 42 and 50 °C). Where necessary, 20 µM hygromycin B, 200 µM viomycin, 900 µM spectinomycin or 20 µM streptomycin were added, and the reaction was monitored at 37 °C.

Arrhenius plot was constructed to calculate the activation energy of translocation. The plot displays the logarithm of a translocation rate constant, ln(k), plotted against the reciprocal of the temperature (1/T). The activation energy of the reaction is defined to be (−R) times the slope of the Arrhenius linear plot, where R is a gas constant with the approximate value of 8.31446 J∙K^−1^∙mol^−1^.

## Figures and Tables

**Figure 1 ijms-22-09614-f001:**
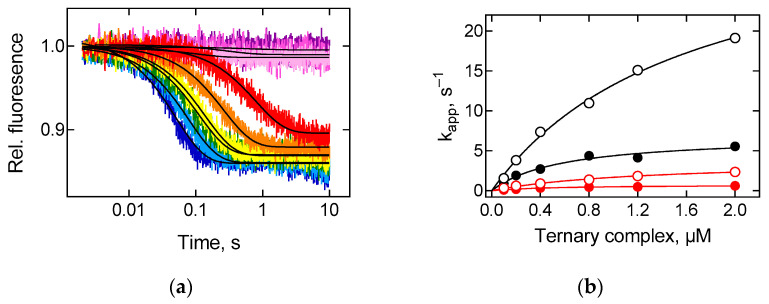
A-site binding kinetics. The ribosomal complexes containing Prf-labelled fMet-tRNA^Met^ in the P site were rapidly mixed with an increasing amount of ternary complex, EF-Tu∙GTP∙Phe-tRNA^Phe^. (**a**) Fluorescence time courses of Prf-labelled ribosomal complex (0.05 µM) interaction with ternary complex (0.1 μM, red; 0.2 μM, orange; 0.4 μM, yellow; 0.8 μM, green; 1.2 μM, blue; 2 μM, dark blue), containing EF-Tu from *E. coli* at 37 °C. The reaction in the presence of inhibitors of A-site binding kirromycin (150 µM, light pink), tetracycline (30 µM, pink) or with buffer TAKM_7_ (violet). Each time course reflects the average of 5–7 technical replicates. Black lines show single-exponential fits. (**b**) Concentration dependence of apparent rate constants (k_app_) on ternary complex (0.1–2 μM). k_app_ values were estimated by the single-exponential fitting of time courses as (**a**) for A-site reactions with EF-Tu from *E. coli* at 20 °C (black circles), at 37 °C (black open circles), EF-Tu from *T. thermophilus* at 20 °C (red circles), at 37 °C (red open circles). Lines show hyperbolic fits, and numerical values are provided in the text. Error bars (s.e.m.) are calculated by the GraphPad Prism software from 5 to 7 technical replicates for each concentration; however, they do not exceed the size of symbols.

**Figure 2 ijms-22-09614-f002:**
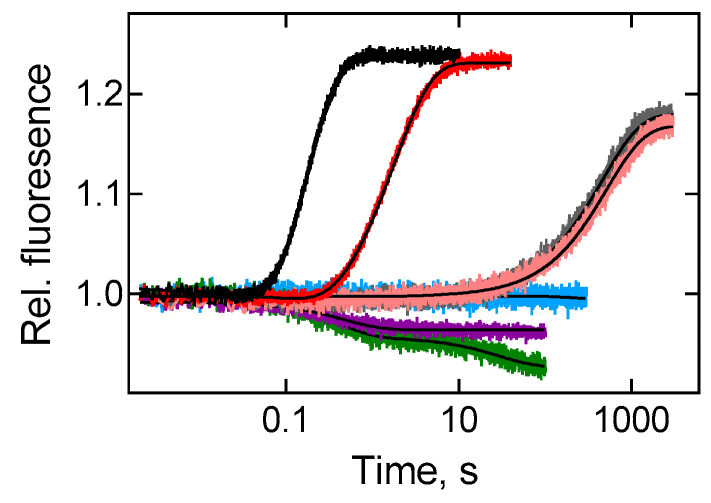
Inhibition of A-site binding kinetics. Time courses of BPY-labelled ribosomal complex (0.1 µM) interaction with ternary complex (1 μM) containing EF-Tu from *E. coli* (black trace) or *T. thermophilus* (red trace) compared to the reaction, inhibited by 30 µM tetracycline with EF-Tu from *E. coli* (grey trace) or *T. thermophilus* (pink trace), 150 µM kirromycin with EF-Tu from *E. coli* (green trace), a modified variant of EF-Tu His84Ala from *E. coli* (violet trace) or buffer TAKM_7_ (blue trace). Each time course reflects the average of 5 to 7 technical replicates. S.e.m. associated with the kinetics were calculated by the GraphPad Prism software.

**Figure 3 ijms-22-09614-f003:**
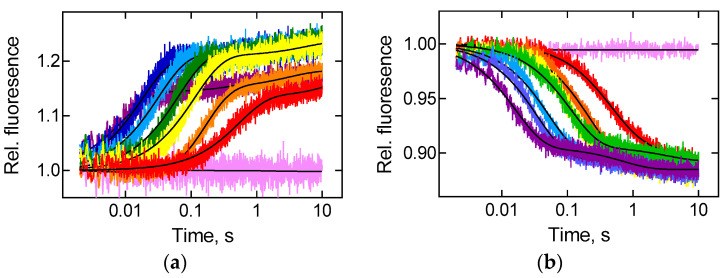
Temperature dependence of tRNA translocation monitored by two fluorescence labels. Time courses of translocation upon rapid mixing of pretranslocation ribosomal complex (0.06 μM) containing deacylated tRNA^Met^ in the P site and fMet-Phe-tRNA^Phe^ in the A site with EF-G (5 μM) from *T. thermophilus* at 15 °C (red), 20 °C (orange), 25 °C (yellow), 30 °C (green), 37 °C (blue), 42 °C (dark blue), 50 °C (violet) or with buffer TAKM_7_ at 20 °C (pink). (**a**) Fluorescence intensity change upon movement of fMet-Phe-tRNA^Phe^(Prf16/17) labelled at the elbow region of tRNA. (**b**) Fluorescence intensity change upon movement of BPY-Met-Phe-tRNA^Phe^ labelled at CCA-end of tRNA. Double-exponential fits are black lines, and numerical values are provided in Figure 4 and Table 1. Each time course reflects the average of 5 to 7 technical replicates. S.e.m. associated with the kinetics were calculated by the GraphPad Prism software.

**Figure 4 ijms-22-09614-f004:**
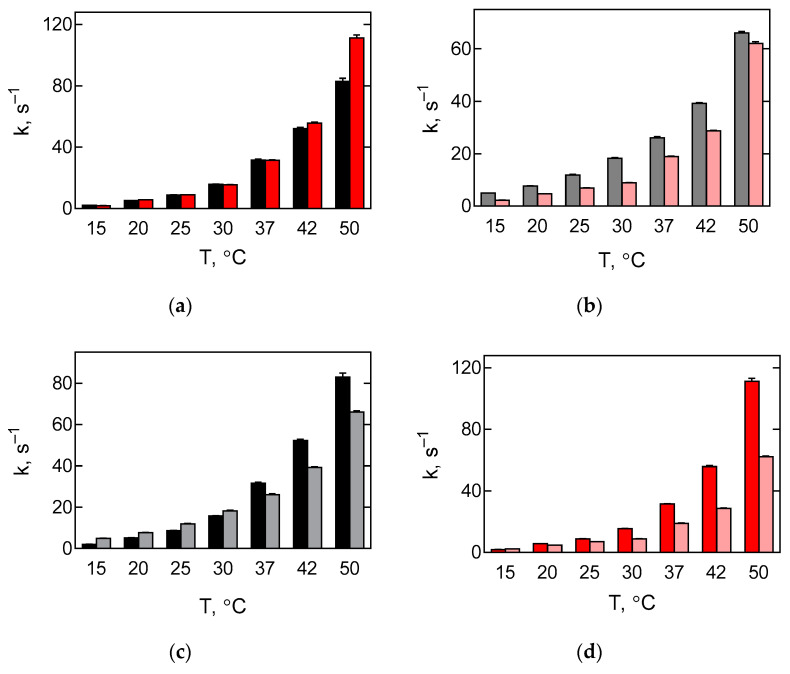
Comparative analysis of temperature dependence of translocation catalysed by EF-G from *E. coli* or *T. thermophilus*. Bar graphs showing rates of translocation (k_1_ of double-exponential fit, Figure 3 and Table 1). Error bars correspond to respective s.e.m. (**a**) Rate of translocation induced by EF-G from *E. coli* (black) or *T. thermophilus* (red) and detected by the label at the elbow region of tRNA. (**b**) Rate of translocation induced by EF-G from *E. coli* (grey) or *T. thermophilus* (pink) and detected by the label at CCA-end of tRNA. (**c**) Rate of translocation induced by EF-G from *E. coli* and detected by the label at the elbow region (black) and CCA-end (grey) of tRNA. (**d**) Rate of translocation induced by EF-G from *T. thermophilus* and detected by the label at the elbow region (red) and CCA-end (pink) of tRNA.

**Figure 5 ijms-22-09614-f005:**
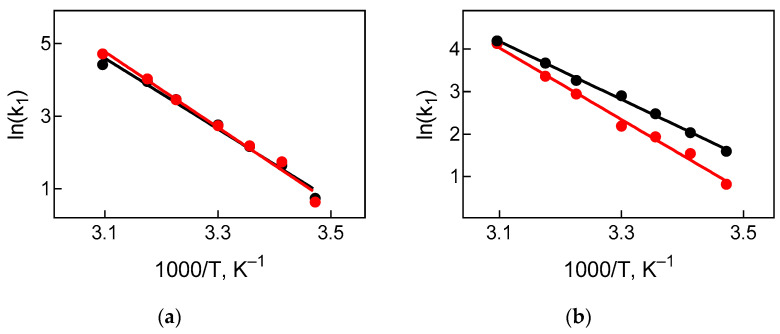
Temperature dependence of rate constants of tRNA translocation (Arrhenius plot). Temperature dependence of translocation rate constants, k_1_, upon addition of EF-G (5 µM) from *E. coli* (black circles) or *T. thermophilus* (red circles) detected by the label (**a**) at the elbow region of tRNA or (**b**) at CCA-end of tRNA. Rate constants are from Table 1. Mean and s.e.m. are plotted; the latter, however, do not exceed the size of symbols. The activation energy of the reaction is defined to be (−R) times the slope of the Arrhenius linear plot, where R is a gas constant with an approximate value of 8.31446 J∙K^−1^∙mol^−1^.

**Figure 6 ijms-22-09614-f006:**
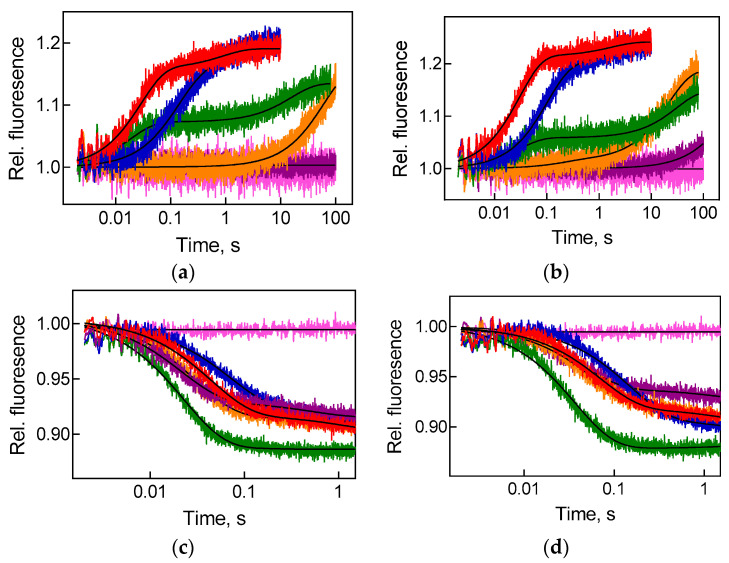
Inhibition of translocation by antibiotics. Time courses of translocation upon rapid mixing of pretranslocation ribosomal complex (0.06 μM) containing deacylated tRNA^Met^ in the P site and fMet-Phe-tRNA^Phe^ in the A site with EF-G (5 μM) from (**a**,**c**) *E. coli* or (**b**,**d**) *T. thermophilus*. (**a**,**b**) Fluorescence intensity change upon movement of fMet-Phe-tRNA^Phe^(Prf16/17) labelled at the elbow region of tRNA. (**c**,**d**) Fluorescence intensity change upon movement of BPY-Met-Phe-tRNA^Phe^ labelled at CCA-end of tRNA. Colour code: reaction without antibiotic (red trace), with streptomycin (dark blue trace), viomycin (violet trace), hygromycin B (orange trace), spectinomycin (green trace) or upon addition of buffer TAKM_7_ instead of EF-G (pink trace). Single- or double-exponential fits are black lines, numerical values are provided in Table 2. Each time course reflects the average of 5 to 7 technical replicates. S.e.m. associated with the kinetics were calculated by the GraphPad Prism software.

**Table 1 ijms-22-09614-t001:** Temperature dependence of rate constants of tRNA translocation.

	*E. coli*		*T. thermophilus*	
T, °C	k_elbow_, s^−1^	k_cca_, s^−1^	k_elbow_, s^−1^	k_cca_, s^−1^
15	2.1	4.9	1.9	2.3
20	5.2	7.7	5.7	4.7
25	8.7	11.9	8.9	6.9
30	15.7	18.2	15.5	8.9
37	31.7	26.1	31.5	19.0
42	52.2	39.2	55.8	28.7
50	83.0	66.1	111.0	62.1

Rate constants determined at 5 µM EF-G; Standard error of the mean did not exceed ±5%; k_elbow_—rate constant of translocation detected by the label at the elbow region of tRNA; k_cca_—rate constant of translocation detected by the label at CCA-end of tRNA.

**Table 2 ijms-22-09614-t002:** Influence of antibiotics on tRNA translocation.

	*E. coli*			*T. thermophilus*		
A/b	k_1 elbow_, s^−1^	k_2 elbow_, s^−1^	k_cca_, s^−1^	k_1 elbow_, s^−1^	k_2 elbow_, s^−1^	k_cca_, s^−1^
No a/b	35	1	23	33	0.6	14
Str	8	1	15	10	0.6	7
Vio	-	-	47	0.004	-	22
Hyg B	0.02	-	33	0.05	-	18
Spc	53	0.07	44	33	0.03	27

Rate constants determined at 5 µM EF-G. No influence of antibiotics on k_2_ of CCA movement was detected. Standard error of the mean did not exceed ±8%. A/b—antibiotic; Str—streptomycin; Vio—viomycin; Hyg B—hygromycin B; Spc—spectinomycin.

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
