# Peer review of "Differential Contribution of Protein Factors and 70S Ribosome to Elongation"

_ijms, 2021, doi:10.3390/ijms22179614_

Round 1

Reviewer 1 Report

Palevska et al examine differences in tRNA binding and translocation between mesophilic and thermophilic elongation factors. They find that EF-Tus act differently while EF-Gs act similarly. The manuscript is well written overall with only minor English language weaknesses. The scope of the work is limited. No further insights are provided about potential molecular differences between the mesophilic and thermophilic EF-Tus that could rationalize the observed differences in tRNA dynamics.

Minor comments:

Methods, Protein expression

The source of the DNA used for cloning EF-Tu and EF-G should be given. Was this synthesized or was it obtain from collaborators? A protein ID should also be provided for these proteins.

L 387: at least a reference (or the methods) needs to be given for the protocols used to aminoacylate and formylate the tRNA

Typos:

Line 119: two dots

Strange symbol in line 364 (should probably be ‘or’)

Author Response

************************************************************************************

Response to Reviewer #1

************************************************************************************

Comments and Suggestions for Authors:

Paleskava et al examine differences in tRNA binding and translocation between mesophilic and thermophilic elongation factors. They find that EF-Tus act differently while EF-Gs act similarly. The manuscript is well written overall with only minor English language weaknesses. The scope of the work is limited. No further insights are provided about potential molecular differences between the mesophilic and thermophilic EF-Tus that could rationalize the observed differences in tRNA dynamics.

Response:       We are thankful to the reviewer for the critical comments and careful reading of the manuscript. Following the suggestion of the reviewer, we revised English throughout the manuscript. We agree with the reviewer that the manuscript does not contain any data that could rationalise the observed differences in tRNA dynamics upon the usage of the mesophilic and thermophilic EF-Tus, as it was beyond the scope of the study. We only mention potential reasons for the observation in the last paragraph. We believe that similarity in EF-Gs action is of more interest for specialists working in the translation field, as EF-G-catalysed translocation is a long-standing mystery with a lot of points to discover.     

Minor comments:

Methods, Protein expression

The source of the DNA used for cloning EF-Tu and EF-G should be given. Was this synthesized or was it obtain from collaborators? A protein ID should also be provided for these proteins.

Response:       We have corrected the text in the Materials and Methods section to state clearly the source of the plasmid DNA used for proteins’ expression as well as the source of the DNA used for cloning. Protein IDs are provided for all proteins.  

L 387: at least a reference (or the methods) needs to be given for the protocols used to aminoacylate and formylate the tRNA

Response:       The reference containing detailed protocols of aminoacylation and formylation of the initiator tRNA was inserted into an appropriate place (Milon, P.; Konevega, A.L.; Peske, F.; Fabbretti, A.; Gualerzi, C.O.; Rodnina, M. V Transient kinetics, fluorescence, and FRET in studies of initiation of translation in bacteria. Methods Enzymol. 2007, 430, 1–30, doi:10.1016/S0076-6879(07)30001-3).  

Typos:

Line 119: two dots

Response:       Extra dot was removed.  

Strange symbol in line 364 (should probably be ‘or’)

Response:       The symbol was removed.  

Reviewer 2 Report

In this article titled “Differential Contribution of Protein Factors and 70S Ribosome to Elongation” the authors have attempted to decipher the effect of heterologous elongation factors from T. thermophilus on E.coli translation steps in vitro.

Major concerns:

  1. Provide statistical analyses for all the figures with n values, P-values, etc. Mention details in the methods section how the studies were done.
  2. Proper negative controls are missing in Figure 1. Also, what does “(0.1-2 μM, traces from red to dark blue)” mean? Provide a clear indication of what these different colors mean? Write detailed figure legends which explain the figure
  3. Proper controls are missing for figure 2. Also, provide statistical analyses.
  4. Figure 3: same criticism as above
  5. Figure 4: Provide statistical analyses and give details in the figure legend/methods section. How were the values obtained?
  6. Figure 5: same as above
  7. Figure 6: same as above

Minor concerns:

  1. The manuscript needs to be thoroughly revised for English grammatical errors and sentence structuring issues present throughout the manuscript. This is especially true for the abstract, first part of the introduction section, lines 93-94 (…detect any amendment occurred…), etc.
  2. the term ‘velocity’ in the abstract seems odd. Maybe the authors can consider more appropriate words like ‘kinetics’?
  3. What is the rationale behind selecting these specific antibiotics concentrations. If these are based on previous studies, provide appropriate references in the methods section.
  4. Typographical error at line 364 “pET24a и pET3a”

Author Response

************************************************************************************

Response to Reviewer #2

************************************************************************************

Comments and Suggestions for the Authors:

In this article titled “Differential Contribution of Protein Factors and 70S Ribosome to Elongation” the authors have attempted to decipher the effect of heterologous elongation factors from T. thermophilus on E.coli translation steps in vitro

Response:       We are thankful to the reviewer for the critical comments and careful reading of the manuscript.

Major concerns:

  1. Provide statistical analyses for all the figures with n values, P-values, etc. Mention details in the methods section how the studies were done.

Response:       Every kinetic trace presented or used for kapp or kav calculation represents the average of 5-7 individual transients recording ≥2000 kinetic points. The averaged traces were evaluated by fitting to a single- or double-exponential function. Average kapp values, kav values and s.e.m. were obtained using Prism 6.02 software (GraphPad Software, USA). S.e.m. are plotted on figures 1, 4, 5 and S1, but can be visible only on figure 4 (bar graph presentation), as the respective s.e.m. do not exceed the size of symbols. All experiments were repeated 2-4 times. The provided statistical analysis is commonly accepted in the field. The detailed description of all experimental procedures was incorporated in the Materials and Methods section and partially introduced into figure legends for clarity.

  1. Proper negative controls are missing in Figure 1. Also, what does “(0.1-2 μM, traces from red to dark blue)” mean? Provide a clear indication of what these different colors mean? Write detailed figure legends which explain the figure

Response:       Following the suggestion of the reviewer, we have introduced negative controls in Figure 1 (2 inhibitors of the A-site binding rection and buffer). We have not done that initially not to overload the figure, as this fluorescence signal is well characterised in the literature. However, we agree that the corrected version of Figure 1 looks better. We are thankful to the reviewer for this improvement. Each color used in panel A represents fluorescence time trace obtained upon the interaction of specific concentration of the ternary complex with the fluorescently labeled initiation complex of fixed concentration. The figure legend was edited to provide detailed information on the experiment and the figure.

  1. Proper controls are missing for figure 2. Also, provide statistical analyses.

Response:       Figure 2 depicts kinetic inhibition of the A-site reaction by tetracycline. As negative controls, the figure contains traces of the reactions inhibited by kirromycin and by deficient in the GTP-hydrolysis variant of EF-Tu H84A. We have additionally introduced a negative control with buffer instead of the ternary complex. As a positive control for the figure, one may consider Figure S1. Figure S1 shows the concentration dependence of the A-site binding monitored by BPY-fluorescence on the ternary complex. Comparison of the values of the rates at the saturation level obtained using Prf20 and BPY labels unambiguously suggests that BPY-fluorescence represents the process of accommodation. Please see our response to critical point 1 by this reviewer for information concerning statistical analyses.

  1. Figure 3: same criticism as above

Response:       We have introduced negative control with buffer instead of EF-G. As additional negative controls, one may consider Figure 6 that contains traces with 4 inhibitors of translocation.

  1. Figure 4: Provide statistical analyses and give details in the figure legend/methods section. How were the values obtained?

Response:       Bar graphs show the rates of translocation (k1 of double-exponential fit, Figure 3 and Table 1). Error bars plotted on the graph correspond to respective s.e.m. The information was added to the figure legend.

  1. Figure 5: same as above

Response:       The graphs show the temperature dependence of the rates of translocation (k1 of double-exponential fit, Figure 3 and Table 1). Error bars plotted on the graph correspond to respective s.e.m., however, do not exceed the size of symbols. The source of the values and the way of calculation of the activation energy introduced into the figure legend.

  1. Figure 6: same as above

Response:       We have introduced negative control with buffer instead of EF-G.

Minor concerns:

  1. The manuscript needs to be thoroughly revised for English grammatical errors and sentence structuring issues present throughout the manuscript. This is especially true for the abstract, first part of the introduction section, lines 93-94 (…detect any amendment occurred…), etc.

Response:       Following the suggestion of the reviewer, we have thoroughly revised the manuscript for English weaknesses. Additional attention was given to the text in the Abstract and the Introduction section.

  1. the term ‘velocity’ in the abstract seems odd. Maybe the authors can consider more appropriate words like ‘kinetics’?

Response:      We have changed the term ‘velocity’ in the Abstract to ‘kinetics’.

  1. What is the rationale behind selecting these specific antibiotics concentrations. If these are based on previous studies, provide appropriate references in the methods section.

Response:      These antibiotics concentrations were chosen based on previous studies, appropriate references were incorporated in the Materials and Methods section.

  1. Typographical error at line 364 “pET24a и pET3a”

Response:       The typographical error was corrected.  

Round 2

Reviewer 2 Report

I would like to congratulate the authors for successfully addressing the concerns raised by me on the previous version of the manuscript.